# Gene Expression Analysis Reveals Potential Regulatory Factors Response to Temperature Stress in *Bemisia tabaci* Mediterranean

**DOI:** 10.3390/genes14051013

**Published:** 2023-04-29

**Authors:** Xiao-Na Shen, Xiao-Di Wang, Fang-Hao Wan, Zhi-Chuang Lü, Wan-Xue Liu

**Affiliations:** 1Department of Basic Medicine, Changzhi Medical College, Changzhi 046000, China; 82101181183@caas.cn; 2State Key Laboratory for Biology of Plant Diseases and Insect Pests, Institute of Plant Protection, Chinese Academy of Agricultural Sciences, Beijing 100193, China; 3Agricultural Genome Institute at Shenzhen, Chinese Academy of Agricultural Sciences, Shenzhen 518120, China

**Keywords:** invasive insects, temperature stress, response mechanism, glucuronidation pathway, *Bemisia tabaci*

## Abstract

Exposure to extreme temperatures can hinder the development of insects and even reduce their survival rate. However, the invasive species *Bemisia tabaci* exhibits an impressive response to different temperatures. This study aims to identify important transcriptional changes of *B. tabaci* occupying different temperature habitats by performing RNA sequencing on populations originating from three regions of China. The results showed that the gene expression of *B. tabaci* populations inhabiting regions with different temperatures was altered and identified 23 potential candidate genes that respond to temperature stress. Furthermore, three potential regulatory factors’ (the glucuronidation pathway, alternative splicing, and changes in the chromatin structure) response to different environmental temperatures were identified. Among these, the glucuronidation pathway is a notable regulatory pathway. A total of 12 UDP-glucuronosyltransferase genes were found in the transcriptome database of *B. tabaci* obtained in this study. The results of DEGs analysis suggest that UDP-glucuronosyltransferases with a signal peptide may help *B. tabaci* resist temperature stress by sensing external signals, such as *BtUGT2C1* and *BtUGT2B13*, which are particularly important in responding to temperature changes. These results will provide a valuable baseline for further research on the thermoregulatory mechanisms of *B. tabaci* that contributes to its ability to effectively colonize regions with considerable temperature differences.

## 1. Introduction

The *Bemisia tabaci* Mediterranean (MED) cryptic species, or Q-biotype whitefly, is an invasive and destructive pest, which originated from the Mediterranean Sea [1,2]. It was first detected in China in 2003 [3]. So far, *B. tabaci* MED has rapidly occupied most provinces in China due to its ability to respond quickly to different temperature conditions in the short term [4,5], showing strong invasiveness and the presence of unique thermoregulation mechanisms. However, these temperature response mechanisms are not common in all *B. tabaci* cryptic species. For example, the native species of *B. tabaci* (AsiaII3) has existed in China for much longer than the *B. tabaci* MED, but its distribution is still limited to selected areas in southern China [6,7]. Previous studies have shown that the temperature tolerance of different geographical populations of *B. tabaci* MED differs notably [8]. Additionally, the temperature tolerance is stable for several subsequent generations by breeding *B. tabaci* populations from different regions for successive generations [8], indicating that the impressive temperature response mechanisms of *B. tabaci* MED are unlikely to have been achieved by simple temperature domestication. Furthermore, previous studies have revealed that some genes in *B. tabaci* MED undergo significant changes in their expression under short-term extreme temperature stress [9]. These findings indicate some specific genes help *B. tabaci* MED to deal with temperature stress. However, the specific regulatory mechanisms’ responses to temperature in *B. tabaci* MED are still unclear.

RNA-Seq can be applied to the study of insect responses to extreme temperatures [10]. We can not only uncover novel genes, but also comprehensively understand the complexity and common characteristics of insect temperature responses by applying high-throughput sequencing to insects. Multiple transcriptome data indicate that insects cope with thermal damage through energy metabolism, protein turnover, detoxification, and stress signal transduction, including response genes such as cytochrome P450, HSPs, adenosine triphosphatases, and glutathione S-transferases [11,12,13,14]. Upregulated immune responses were helpful in dealing with cold stress in *Tenebrio molitor* and *Ostrinia furnacalis* [13,15]. Furthermore, cuticular proteins are modulated by exposure to cold stress in *Liriomyza trifolii* and sea buckthorn moth [16,17] and play a role in the tolerance to coldness in New Zealand stick insects [18]. Although there are transcriptome data associated with temperature stress in insects, the number of differentially expressed genes (DEGs) and their levels of upregulation vary in different insects, which may be related to transient stress and species.

Therefore, it is necessary to investigate the mechanisms of temperature response under natural temperature stress from a specific species. Studying natural populations can provide mechanistic insights that accurately reflect specific response mechanisms of a species in the wild. Additionally, a previous study reported that a *B. tabaci* field population in Turpan exhibited a stronger heat response, whereas that in Harbin exhibited a stronger cold response [8], which also suggests the necessity of studying natural populations. In this study, we captured *B. tabaci* MED from Haikou (N), Turpan (T), and Harbin (H) City in China with distinct temperatures and used high-throughput transcriptome sequencing of three geographically distinct populations of *B. tabaci* MED from regions with different environmental temperatures to identify key transcriptional differences. This work can help us understand the response mechanisms to the environmental temperature of invasive species when they occupy different habitats by exploring transcriptional responses to temperature stress in natural populations.

## 2. Materials and Methods

### 2.1. Experimental Insects

In 2015, the highest daily average temperature during the summer was 103 °F (43 °C) in Turpan City, 93 °F (32.7 °C) in Haikou City, and 81 °F (28.9 °C) in Harbin City. The daily average temperature information was extracted from ©WeatherSpark.com on 1 July 2022 and is shown in Figure 1. Three *B. tabaci* MED field populations were obtained from Haikou City (Hainan Province; 20.01° N, 110.34° E), Turpan (Xinjiang Province; 45.56° N, 89.13° E), and Harbin (Heilongjiang Province; 42.93° N, 126.70° E) in China in July 2015. In these cities, the highest temperature was recorded in July (Appendix A). There are significant differences in the types of crops grown locally in different regions of China, and the main types of crops harmed by *B. tabaci* MED are also different. Therefore, when collecting *B. tabaci* MED field populations, we try to choose common hosts, such as cotton and tomato plants, and collect as many adults as possible. The collection process in each region is completed within 3 h. Subsequent molecular identification was conducted by interrogation of the cytochrome oxidase 1 gene. Field populations were reared on cotton plants grown at 24–26 °C and 50–60% humidity with a 14:10 h light: dark cycle. Adults from each population are divided into three parts and kept in three insect cages, forming three biological replicates. After being reared for two generations, 200 adults of *B. tabaci* MED from each geographical population were collected within 72 h of feathering and immediately frozen in liquid nitrogen for 2 min and stored at −80 °C until RNA extraction. Randomly sampled from the above each treatment. The purpose of feeding captured *B. tabaci* adults under common laboratory conditions for two generations was to overcome potential maternal effects and eliminate any influence of natural host plants [19]. Previous studies have shown that the thermal knockdown time and the recovery time from these regions after low-temperature knockdown are not affected by rearing two consecutive generations under common laboratory conditions [8].

### 2.2. RNA Extraction

Two hundred *B. tabaci* adults in each treatment were subjected to RNA extraction. Total RNA was extracted using TRIzol reagent (Invitrogen, Cat. No. 15596026, Carlsbad, CA, USA). The RNA quality was checked by electrophoresis on 1% agarose gels and measuring optical density values using a NanoPhotometer^®^ spectrophotometer (IMPLEN, Munich, Germany). RNA concentration was measured using a Qubit^®^ RNA Assay Kit on a Qubit^®^ 2.0 Fluorometer (Life Technologies, Carlsbad, CA, USA). RNA integrity was assessed using an RNA Nano 6000 Assay Kit on an Agilent Bioanalyzer 2100 system (Agilent Technologies, Santa Clara, CA, USA).

### 2.3. Library Construction

For each sample, 1.5 μg of RNA was used as input material for mRNA purification. RNA sequencing (RNA-Seq) libraries were generated using the NEBNext^®^ UltraTM RNA Library Prep Kit for Illumina^®^ (NEB, Ipswich, MA, USA) following the manufacturer’s recommendations, and index codes were added to attribute sequences to specific samples. Briefly, mRNA was purified from total RNA using poly T oligo-coated magnetic beads. Fragmentation was conducted using divalent cations under elevated temperatures in NEBNext First-Strand Synthesis Reaction Buffer (5X). First-strand cDNA was synthesized using random hexamer primers and M-MuLV reverse transcriptase (RNase H^−^). Second-strand cDNA synthesis was subsequently performed using DNA Polymerase I and RNase H. The remaining overhangs were converted to blunt ends via exonuclease/polymerase activity. The 3′ ends of the DNA fragments were adenylated and ligated to NEBNext adapters with a hairpin loop structure to prepare for hybridization. Library fragments with preferential sizes of 150–200 bp were purified using the AMPure XP system (Beckman Coulter, Beverly, USA). Then, for each sample, 3 μL USER Enzyme (NEB, Ipswich, MA, USA) was mixed with size-selected, adaptor-ligated cDNA and incubated at 37 °C for 15 min, followed by 5 min at 95 °C before polymerase chain reaction (PCR). PCR was performed using the Phusion High-Fidelity DNA polymerase, universal PCR primers, and index (X) primers. The final PCR products were purified using the AMPure XP system, and library quality was assessed using an Agilent Bioanalyzer 2100 system.

### 2.4. Transcriptome Assembly and Gene Functional Annotation

The sequenced reads (raw reads) contained some reads with adapters and low quality (the alkali base of Q_phred_ ≤ 20 accounts for over 50% of the total read length.). To ensure the quality of information analysis, the above reads were filtered to obtain clean reads. Clean reads were assembled using Trinity (version 2.1.1) to obtain a high-quality unigene library. The databases of Nr (NCBI non-redundant protein sequences: https://ftp.ncbi.nlm.nih.gov/blast/db/FASTA/, accessed on 11 March 2017), Nt (NCBI non-redundant nucleotide sequences: https://ftp.ncbi.nih.gov/blast/db/FASTA/, accessed on 11 March 2017), Pfam (Protein family: http://pfam.xfam.org/, accessed on 19 April 2023), COG (Clusters of Orthologous Groups of proteins: http://www.ncbi.nlm.nih.gov/COG/, accessed on 16 March 2017), Swiss-Prot (a manually annotated and reviewed protein sequence database: http://www.gpmaw.com/html/swiss-prot.html, accessed on 20 March 2017), KO (KEGG Ortholog database: http://www.genome.jp/kegg/ko.html, accessed on 20 March 2017), and GO (Gene Ontology: http://geneontology.org, accessed on 20 March 2017) were used to annotate all assembled unigenes using the BLAST algorithm with an E-value cut-off of 10^−5^ [20].

### 2.5. Alternative Splicing Analysis

We used rMATS software (http://rnaseq-mats.sourceforge.net/index.html, accessed on 1 April 2017) to perform an alternative splicing analysis on our RNA-Seq data. Taking each comparison for differential variable splicing analysis as a unit, we first counted the types and quantities of variable splicing events that occurred and then calculated the expression levels of each type of variable splicing event. Shearing events were also analyzed to determine differences between populations. In the quantitative process, rMATS adopts two quantitative methods: (1) junction count only and (2) reads on target and junction counts.

### 2.6. Differential Gene Expression and Functional Enrichment Analysis

In this study, the HTSeq software was used to analyze gene expression in each sample using the union model. The correlation of gene expression levels between samples is an important indicator of the reliability of experiments; correlation coefficients close to 1 indicate a high similarity of expression patterns between samples. In this study, we required that the correlation coefficient between biological replicates be greater than 0.8. The expected number of fragments per kilobase of transcript sequence per million base pair sequences (FPKM) is used to estimate expression levels [21]. For repeated samples under the same experimental conditions, the final FPKM is the average of all repeated data. After calculating the expression level of each unigene, a differential expression analysis was performed between all pairs of samples using the DESeq R package (1.10.1). DESeq provides statistical routines to determine differential expression in digital gene expression data using a model based on a negative binomial distribution. The resulting *p*-values were adjusted using Benjamini and Hochberg’s approach to control the false discovery rate. Genes with an adjusted *p*-value < 0.05 were considered differentially expressed.

Gene ontology (GO) enrichment analysis of the differentially expressed genes (DEGs) was implemented using the GOseq R package based on a Wallenius non-central hyper-geometric distribution [22], which can adjust for gene length bias in DEGs. We used KOBAS [23] software to test the statistical enrichment of DEGs in the KEGG database.

### 2.7. Bioinformatics Analysis

Sequence alignment and identity analyses were performed using DNAMAN (version 8.0; Lynnon BioSoft, Quebec, Canada). The functional domains of the deduced protein sequences of target genes were identified using SMART software (http://smart.embl-heidelberg.de/, accessed on 11 June 2022). Signal peptides and cross-membrane domains were predicted using SignalP 5.0 Server (http://www.cbs.dtu.dk/services/SignalP/, accessed on 11 June 2022) and TMHMM Server v.2.0 (http://www.cbs.dtu.dk/services/TMHMM, accessed on 11 June 2022/). Multiple protein sequences were aligned using DNAMAN 8.0 and implemented in the MEGA 7 software package to evaluate the molecular evolutionary relationships between target genes in *B. tabaci* and various other insects. A phylogenetic tree was constructed using the maximum likelihood method in MEGA 7. Bootstrap majority consensus values for 1000 replicates are indicated at each branch point (%).

## 3. Results

### 3.1. Sequence Assembly and Functional Annotation

Transcriptome libraries were constructed using RNA samples extracted from the three field populations of *B. tabaci* and sequenced using an Illumina HiSeq 2500 instrument in triplicate. The total number of clean reads obtained from each library was 44.58 ± 2.46 (mean ± SD) million in Haikou samples, 34.90 ± 8.80 million in Turpan samples, and 50.08 ± 4.95 million in Harbin samples (Appendix A). Then, we aligned the filtered sequences using HISAT software and found that approximately 60% of the reads uniquely mapped to the reference genome [24], while only no more than 10% are mapped to multiple regions of the genome (Appendix A). These results indicate that the experimental samples were free from contamination from other organisms and that the reference genome used was suitable for this study. Regarding read distribution in the genome, approximately half of the reads were assigned to intergenic regions, due to incomplete genome annotation (Appendix A).

### 3.2. Alternative Splicing

Our analysis revealed the occurrence of temperature-responsive alternative splicing events responsive to temperature in *B. tabaci*, which comprised skipped exons (SE) and mutually exclusive exons (MXE) (Appendix A). Over 2800 SE events were detected among the three populations, of which 185 events were identified as significantly different between the Turpan and the Harbin populations (T vs. H), and 159 events were identified as significantly different between Haikou City and the Harbin (N vs. H) populations; these significantly different events (Sig events) included up- and downregulated events. Furthermore, more than 300 MXE events were detected, of which approximately 50 to 80 were significantly different. The numbers of total alternative splicing events and Sig events for the two comparisons (T vs. H and N vs. H) are shown in Appendix A. These results indicate the importance of alternative splicing in responses to environmental stresses in invertebrates with relatively simple genetics and body structures.

### 3.3. Expression Differences between High- and Low-Temperature Populations

RNA-seq was used to determine the transcriptome of *B. tabaci* MED cryptic species from three geographical populations, and the two-dimensional hierarchical clustering was performed using the “heatmap” package in R to analyze 228 DEGs (Figure 1) [20]. Then, clustering the samples according to the expression of DEGs divided them into two clusters, with the two high-temperature field populations preferentially clustering together (Figure 2). These data suggested a high transcriptional similarity between the two high-temperature field populations. Furthermore, this finding also supported the reliability of our experimental design, as temperature response outweighed the influence of other environmental factors in the clustering analysis.

To uncover transcriptional expression patterns of DEGs in *B. tabaci* in response to temperature stress conditions, the list of DEGs was submitted to http://bioinfogp.cnb.csic.es/tools/venny/index.html on 10 June 2017 for Venn analysis. The results showed that 23 DEGs were shared by the two high-temperature populations (Table 1). Among these DEGs, three types of DEGs caught our attention: the UDP-glucuronosyltransferase (UGT) family, the splicing factor, and the transposable element P transposase.

### 3.4. Functional Annotation of Differentially Expressed Genes

Gene ontology (GO) terms were used to functionally classify the predicted *B. tabaci* proteins of the DEGs according to their molecular functions, biological processes, and cellular components. The analysis results showed that approximately 60% of DEGs were distributed in “Biological Process”, 27% in “Molecular Function”, and only 15% genes in “Cellular Component” (Appendix A), which suggested a strong correlation between the mechanisms of temperature responses and genes involved in various biological processes, including glucuronidation, chromosome organization, segregation, and mRNA splicing.

Then, a KEGG pathway analysis was performed to identify the pathways in which DEGs are involved. Subsequently, by comparing the KEGG analysis results of two high-temperature and low-temperature field populations, it was found that the DEGs were mainly involved in the following pathways: starch and sucrose metabolism, retinol metabolism, porphyrin and chlorophyll metabolism, pentose and glucuronate interconversions, metabolism of xenobiotics by cytochrome P450, and ascorbate and aldarate metabolism (Figure 3). The results indicate that the conversion and metabolism of sugars are important aspects of the response to temperature in *B. tabaci*. In addition, the pathways of cytochrome P450 and UDP-glucanosyltransferase catalyzing oxidation and glucuronidation in lipophilic small molecule metabolism are notable regulatory pathways for temperature responses in *B. tabaci*.

### 3.5. Bioinformatics Analysis of Genes from a Target Regulatory Pathway in B. tabaci

UDP-glucuronosyltransferases, identified by differential gene expression analysis and KEGG analysis, catalyze oxidation and glucuronidation during the metabolism of small lipophilic molecules. In this study, the gene sequences of 12 enzymes from the UGT family were identified from our RNA-Seq data (Section 3.1) and used for bioinformatic analysis. The analysis results showed that the UDP-glucuronosyltransferase gene, *BtUGTs*, in *B. tabaci* MED encodes a 510–540 amino acid protein. However, the amino acid sequences of the 12 enzymes in the UGT family do not have identical structures. Some enzymes contain an N-terminal signal peptide, the Pfam domain (named UDPGT, PF00201), 1–2 transmembrane domains, and the C-terminal domain (Glyc0_tran_28_C) with the UDP-N-acetylglucosamine (UDP-GlcNAc) binding site, while other enzymes lack at least one of these structures. The structural characteristics of UDP-glucuronosyltransferases in *B. tabaci* are shown in Table 2. This result indicates that UDP-glucuronosyltransferases in *B. tabaci* are membrane-bound microsomal enzymes. In addition, UDP-glucuronosyltransferases catalyze the transfer of glucuronic acid to a wide variety of exogenous and endogenous lipophilic substrates, which suggests their importance in eliminating exogenous substances and toxic secondary metabolites [25,26]. In this study, structural analysis of UGT family enzymes detected in *B. tabaci* suggested that UDP-glucuronosyltransferases with a signal peptide contain only one C-terminal transmembrane domain, whereas those without a signal peptide have two transmembrane domains (one at each terminus) (Table 2). These results indicate that some enzymes in the UGT family are responsible for transmembrane transport, and others may be receptors for extracellular signals in response to external stress.

Differential expression analysis also showed that both UDP-glucuronosyltransferase 2B13 (*BtUGT2B13*) and UDP-glucuronosyltransferase 2C1 (*BtUGT2C1*) were overexpressed in two *B. tabaci* high-temperature populations. The sequence structures of *BtUGT2B13* and *BtUGT2C1* were identical, and both contained a signal peptide, a UDPGT domain, a Glyc0_tran_28_C domain, and a single C-terminal transmembrane domain (Figure 4a). However, the transcriptional responses of *BtUGT2B13* and *BtUGT2C1* to temperature were different. *BtUGT2B13* was upregulated and *BtUGT2C1* was downregulated in high-temperature populations. Furthermore, the two genes did not cluster together in our phylogenetic analysis (Figure 4b). These results suggest that both *BtUGT2B13* and *BtUGT2C1* may help *B. tabaci* resist temperature stress by sensing external signals, although by different mechanisms.

Then, through multiple sequence alignment analysis, it was found that the UGT2C1 and UGT2B13 sequences were moderately conserved across multiple species, with several highly conserved regions (Figure 4c,d). This result indicates that these conserved regions are likely to be the main functional domains. Although there are several highly conserved regions between the two sequences of BtUGT2B13 and BtUGT2C1, they exhibited only 23.7% homology (Figure 4e), implying that enzymes of the UGT family likely have similar but not completely redundant functions.

## 4. Discussion

In this study, we used high-throughput transcriptome sequencing of three geographically distinct populations of *B. tabaci* MED from regions with different environmental temperatures to identify key transcriptional differences. By exploring transcriptional responses to temperature stress in natural populations, this work can help to understand the response mechanisms to the environmental temperature of invasive species when they occupy different habitats, which is important in the context of the current climate change.

The results showed that the gene expression of *B. tabaci* populations inhabiting regions with different temperature was altered and identified 23 potential candidate genes that respond to temperature stress. Three types of DEGs caught our attention: the UDP-glucuronosyltransferase (UGT) family, the splicing factor, and the transposable element P transposase. First, splicing factor 27 is required for the activation of pre-mRNA splicing [27] and may have a scaffolding role in the assembly of the spliceosome, as it communicates with all other components of the core complex. Our alternative splicing analysis results demonstrated the occurrence of upregulated splicing events between high-temperature and low-temperature field populations, supporting a role for splicing in the temperature response. Second, previous work has shown that transposable element activation is regulated by chromatin-modifying genes in Drosophila [28,29], and the chromatin remodeling factor, ISWI, has been increasingly recognized as an important factor in the response of *B. tabaci* to temperature stress [30]. Third, the UGT family encodes several UDP-glucuronosyltransferases, which are enzymes of the glucuronidation pathway that transform small lipophilic molecules into water-soluble, excretable metabolites [31]. In this catalytic process, the water solubility of receptor molecules is greatly improved, and the efflux of these molecules from the body is promoted. The UGT family participates in the most important glucuronate metabolic pathway. In plants, UDP-glucuronosyltransferase is involved in growth and development, metabolic regulation, and metabolite synthesis [31,32]; regulation of the metabolism and production of secondary metabolites can defend the organism against adverse conditions. Research indicates that cytochrome P450 and UDP-glucuronosyltransferase catalyze oxidation and glucuronidation in drug metabolism, respectively [33]. Cytochrome P450 plays an important role in the processes that respond to temperature in multiple organisms, including insects [34,35,36,37,38,39,40]. We propose that UDP-glucuronosyltransferase may play a significant role in temperature tolerance by metabolizing harmful substances produced at extreme temperatures, similar to the mechanism of cytochrome P450 activity. In this study, two UDP-glucuronosyltransferases were upregulated in the two high-temperature populations, supporting the importance of glucuronidation in temperature response. In addition, KEGG analysis of DEGs also showed that the pathways of cytochrome P450 and UDP-glucuronosyltransferase catalyzing oxidation and glucuronidation in lipophilic small molecule metabolism are notable regulatory pathways for temperature responses in *B. tabaci*.

UDP-glucuronosyltransferases catalyze oxidation and glucuronidation during the metabolism of small lipophilic molecules. A total of 12 UDP-glucuronosyltransferase genes were found in the transcriptome database of *B. tabaci* obtained in this study. Bioinformatics analysis of 12 genes in UGT family revealed that UDP-glucuronosyltransferases in *B. tabaci* are membrane-bound microsomal enzymes. Additionally, structural analysis of UGT family enzymes detected in *B. tabaci* suggested that some enzymes in the UGT family are responsible for transmembrane transport, and others may be receptors for extracellular signals in response to external stress. Differential expression analysis showed that UDP-glucuronosyltransferases with a signal peptide (*BtUGT2B13* and *BtUGT2C1*) were overexpressed in two *B. tabaci* high-temperature populations. The results further confirmed that a UGT enzyme with a signal peptide may help *B. tabaci* resist temperature stress by sensing external signals.

This study provides some transcriptional regulation information for further research on response mechanisms to temperature from the perspective of alternative splicing and chromatin remodeling. In addition, an important regulatory pathway involving the UGT family response to external stress has been discovered. The function of the UGT family in response to temperature stress in *B. tabaci* still needs to be explored in the future. Studies on the response mechanisms to the environmental temperature in invasive species will help to understand why they continue to invade and colonize low-temperature areas.

## 5. Conclusions

In this study, RNA-Seq was used to unravel the transcriptional landscapes of *B. tabaci* MED isolated from three geographical populations with different temperature ranges (Turpan, Harbin, and Haikou City). Although there are many transcriptome data associated with temperature stress in insects, we found that the number of DEGs and their levels of upregulation varies in different insects, which may be related to limitations of transient stress and species. Therefore, the comparative transcriptomics of natural populations of *B. tabaci* may shed new light on the response mechanisms to temperature underlying the successful invasion of this species into different regions. Our data intuitively show changes in gene expression after the invasion of new environments with different temperatures. This temperature-bound selection pressure also partly explains why a single genetically uniform population of *B. tabaci* dominates farmscapes with relatively uniform temperature patterns [41]. We have identified many potential candidate genes that respond to temperature in *B. tabaci*, involved in the process of alternative splicing and changes in chromatin structure. These results indicate that the response mechanisms to temperature may require complex biological processes that involve numerous changes in gene expression. Three potential regulatory factors (glucuronidation pathway, alternative splicing, and changes in chromatin structure) were differentially expressed at different environmental temperatures. Among these, the glucuronidation pathway is a notable regulatory pathway response to temperature in *B. tabaci*. Some UDP-glucuronosyltransferases catalyze oxidation and glucuronidation during small lipophilic molecule metabolism following the response to external stress signals by other UDP-glucuronosyltransferases with signal peptides, such as *BtUGT2C1* and *BtUGT2B13*. Results in the current study vastly improve contemporary knowledge about *B. tabaci* responses to temperature variations and provide a framework to study the connection between temperature response mechanism and its ability to effectively colonize regions with considerable temperature differences.

## Figures and Tables

**Figure 1 genes-14-01013-f001:**
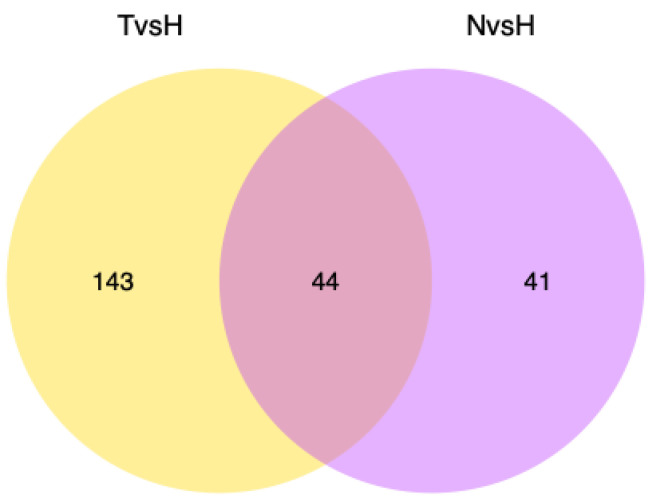
Venn diagram of differentially expressed genes from two comparisons: Turpan vs. Harbin (T vs. H) and Haikou City vs. Harbin (N vs. H).

**Figure 2 genes-14-01013-f002:**
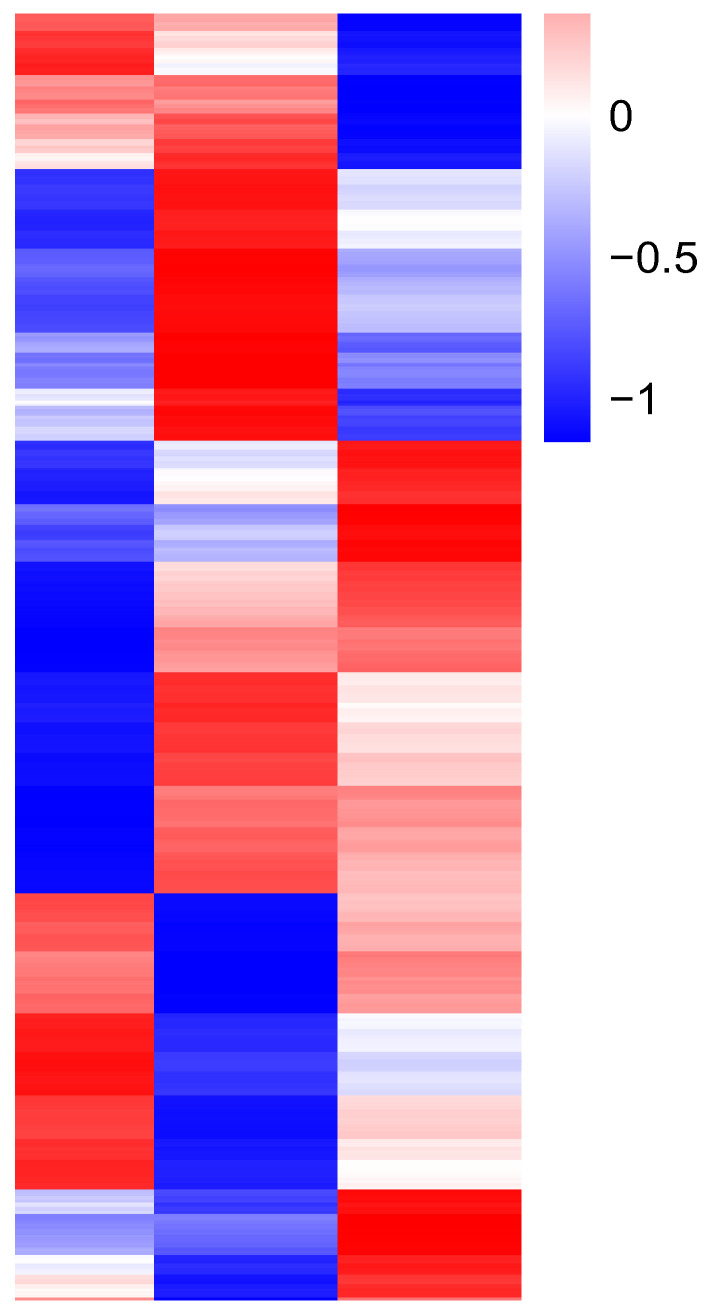
Heatmap clustering analysis of differentially expressed genes in *B. tabaci* MED cryptic species from three geographic populations. The log_10_ (FPKM + 1) value was normalized, scaled, and clustered. Red indicates highly expressed genes, and blue indicates lowly expressed genes. Shading from red to blue indicates log_10_ (FPKM + 1).

**Figure 3 genes-14-01013-f003:**
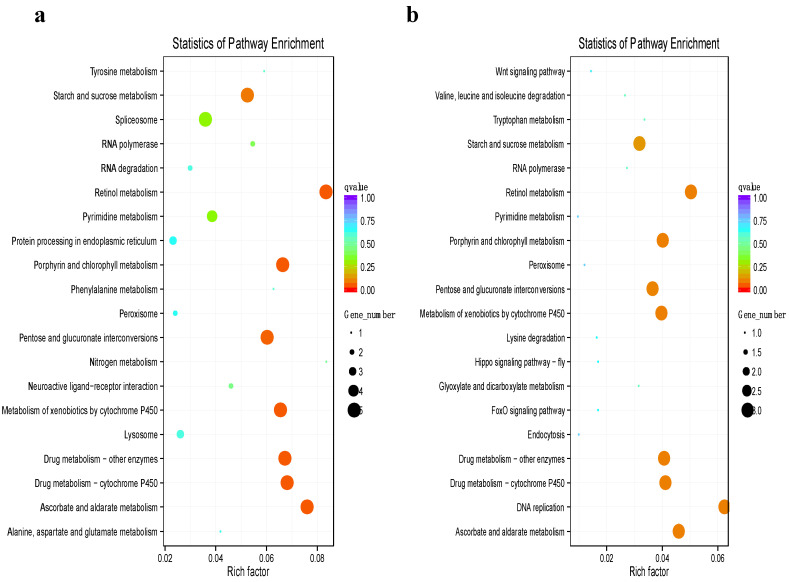
Distribution of KEGG functional groups within differentially expressed transcripts from two comparisons: Turpan vs. Harbin (**a**) and Haikou City vs. Harbin (**b**). The bar chart corresponds to the matched entries of differentially expressed transcripts in their own functional category. The size of the dot indicates the number of differentially expressed genes in the path, and the color of the dot corresponds to different Q value ranges.

**Figure 4 genes-14-01013-f004:**
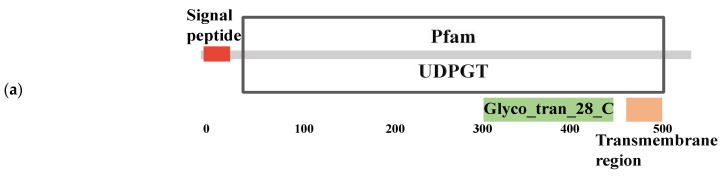
(**a**) Diagram of the functional domains of BtUGT2C1 and BtUGT2B13. (**b**) Phylogenetic analysis based on amino acid sequences of UDP-glucuronosyltransferases in B. tabaci. (**c**,**d**) Multiple alignments of the deduced amino acids of UGT2C1 (**c**) and UGT2B13 (**d**) between species. (**e**) Alignment of the amino acids of BtUGT2C1 and BtUGT2B13 in B. tabaci. Dark color shows highly conservative areas.

**Table 1 genes-14-01013-t001:** Differentially expressed genes shared by high-temperature populations from two comparisons (Turpan vs. Harbin and Haikou City vs. Harbin).

Abbreviated Name	Full Gene Name	Amino Acids	T vs. H	N vs. H	Putative Functions (from NCBI Database)
*BtADRO*	adrenodoxin oxidoreductase	459	up	up	
*BtUraD*	2-oxo-4-hydroxy-4-carboxy-5-ureidoimidazoline decarboxylase	174			
*BtUGT2B13*	UDP-glucuronosyltransferase 2B13	522			
*BtMet16*	U6 small nuclear RNA (adenine-(43)-N(6))-methyltransferase	524			involved in regulation of mRNA splicing via spliceosome
*Btmsps*	Protein mini spindles	1997			
*BtSPF27*	Pre-mRNA-splicing factor SPF27	221			activates pre-mRNA splicing
*BtCacybp*	Calcyclin-binding protein	224			
*BtMal1*	Maltase 1	600			maltose alpha-glucosidase activity
*BtOzf6*	Oocyte zinc finger protein XlCOF6	381			trypsin-like serine protease activity
*BtUbp5*	Ubiquitin carboxyl-terminal hydrolase 5	813			
*BtBOLA3*	BolA-like protein 3	104			
*BtF10C1*	Protein FRA10AC1 homolog	249			involved in dephosphorylation
*BtALG13*	UDP-N-acetylglucosamine transferase and deubiquitinase ALG13	745			involved in proteolysis
*BtSpin*	Protein spinster	531	down	down	enables transmembrane transporter activity
*BtUGT2C1*	UDP-glucuronosyltransferase 2C1	534			
*BtAnnu*	Annulin	756			
*BtSiah2*	E3 ubiquitin-protein ligase siah2	155			involved in apoptotic process
*BtTrePt*	Transposable element P transposase	1157			
*BtKAE1*	tRNA N6-adenosine threonylcarbamoyltransferase	334			involved in chromosome organization, enables chromatin DNA binding

**Table 2 genes-14-01013-t002:** Structural characteristics of UDP-glucuronosyltransferases in *B. tabaci*.

Abbreviated Name	Amino Acids	Signal Peptide	Transmembrane Region	Pfam: UDPGT	Pfam: Glyco_tran_28_C	NCBI Reference Sequence ID
*BtUGT1-8*	533	1–19	474–496	31–517	(-)	XP_018907955.1
*BtUGT1-9*	527	(-)	7–29, 492–514	21–524	(-)	XP_018897720.1
*BtUGT2A1*	528	1–19	390–409	21–527	330–457	XP_018909896.1
*BtUGT2B1*	518	(-)	7–29, 480–502	25–515	303–444	XP_018896780.1
*BtUGT2B2*	519	1–19	473–495	22–514	(-)	XP_018899270.1
*BtUGT2B7*	527	(-)	7–29, 480–508	21–523	(-)	XP_018896827.1
*BtUGT2B13*	522	1–19	479–501	21–515	304–444	XP_018905849.1
*BtUGT2B15*	518	1–22	480–502	24–517	480–502	XP_018896916.1
*BtUGT2B18*	537	(-)	7–29, 494–516	26–534	340–469	XP_018912485.1
*BtUGT2B20*	526	(-)	12–34, 488–510	31–525	(-)	XP_018911460.1
*BtUGT2B31*	512	1–20	480–502	21–504	(-)	XP_018905835.1
*BtUGT2C1*	534	1–26	482–504	29–510	302–446	XP_018901567.1

(-) implies there is no such structure.

## Data Availability

The data presented in this study are openly available in SRA database, reference number PRJNA674400.

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
