# Peer review of "Gene Expression Analysis Reveals Potential Regulatory Factors Response to Temperature Stress in Bemisia tabaci Mediterranean"

_genes, 2023, doi:10.3390/genes14051013_

Round 1

Reviewer 1 Report

The authors should read the article once again to see minor English language improvement. For example, one sentence in the abstract is highlighted. 

Discussion section should be added in the revised version, so that, complete review of the article may be done. 

Author Response

Thank you very much for your contribution to this manuscript. We have revised the manuscript according to your opinion. Details are as follows:

Q1. The authors should read the article once again to see minor English language improvement. For example, one sentence in the abstract is highlighted. 

A1. We have revised the English expressions in the article, including bold, grammar, conjunction, etc

Q2. Discussion section should be added in the revised version, so that, complete review of the article may be done. 

A2. The discussion has been separately extracted from the results and edited as a separate part, as detailed in 4. Discussion.

Reviewer 2 Report

I would suggest to modify the Title. Title is not catchy/attractive.
Abstract: This section seems scary and unclear. I would suggest to materials and methods in this
section. Results presented in this sections should be explain precisely with addition of numbers.
Introduction: Introduction: Why study was conducted on this crop? The significant novel point of the
study over the precedent studies is not clear. This part can be shortened and try to be specific,
irrelevant information can be deleted. There are two major concerns with this MS. First one is
grammatical mistakes, language errors, and typographical mistakes. Second, the authors did not conceive
a strong idea from the review literature. Paragraphs and sentences did not have any links. Materials and methods: How many replications per treatment? and where the data collected? its better if collected data from various regions?. Results and Discussion: In results, there is a striking lack of connectors between sentences and
leading to confusion. There are many values in results that increase the ubiquity in results. I would suggest presenting your results by increasing/decreasing %age. One way to improve the Discussion is
to avoid repeating results in this part. Discussion is very shallow and needs in-depth discussion
with the recent literature published. In discussion, there is a lack of a mechanistic approach.

Reviewer 3 Report

My review of this manuscript comes from the perspective of a physiological ecologist and not that of a molecular biologist, therefore, I cannot comment on the specifics of techniques and methodology. I can say that the objectives and experimental design were sound and the results were, at least in my opinion, fascinating. Of particular note, linking the DEGs to specific physiological function associated with heat response was especially interesting and a welcome change from papers that just document changes without exploring the implications of such changes.

There are some minor issues with word choice and formatting in the mss, which I highlighted in my copy of the mss. The legend and completed Figure 1 are missing from my copy, and Figure 2, the Venn diagram, is incomplete. Beyond these issues, I think the manuscript is very clean and close to being suitable for publication.

Author Response

Thank you very much for your contribution to this manuscript. We have revised the manuscript according to your opinion. Details are as follows:

Q1. There are some minor issues with word choice and formatting in the mss, which I highlighted in my copy of the mss.

A1. Thank you very much for your contribution to this manuscript again. We have revised them based on the revised version you provided.

Q2. The legend and completed Figure 1 are missing from my copy, and Figure 2, the Venn diagram, is incomplete. Beyond these issues, I think the manuscript is very clean and close to being suitable for publication.

A2. Perhaps due to version issues, these parts are missing. We have reinserted the images and hope to resolve this issue. Alternatively, we can provide original images to solve the problem.

Reviewer 4 Report

The study entitled “Gene Expression Analysis Reveals Potential Response Factors to Temperature in Bemisia tabaci Mediterranean” gives preliminary results on the genes and pathways involved in temperature stress of Bemisia tabaci. The authors performed RNA-seq analysis in three populations originated from different areas in China and identified changes in the gene expression. Based on their results, they suggest that specific genes and regulatory factors might be involved in the response of Bemisia tabaci in different temperatures.

The main flaw of this study is the experimental design. The authors performed tha analysis on samples collected from the nature which they reared in the lab for two generations. They did not do any analysis on the initial wild material and they did not use and control (e.g. a long-adapted lab population) that would serve as a baseline for their analysis. 

As a result, the reader cannot be sure if the data presented are due to temperature changes, lab artificial rearing, bottleneck events, or all of these together. I suggest a major revision through a different perspective for this study.

L57-59: a reference is required for this statement

L61: this sentence should be rephrased to: “RNA-Seq can be applied to the investigation…”

L75-84: why is this part in bold?

L86-103: this part lacks clarity and the authors should give more information to make it clearer to the reader. The collection protocol is not clearly stated. How were the insects collected? Did they use traps or directly from the host? If so, which were the host plants? How many days did the collection last and what were the temperature and weather conditions on that day? How many insects were collected per area? In addition, why was July chosen as the month of collection? Did the authors have data from previous years that showed that July is in general the hottest month in these areas? The explanation of the two-generation rearing is not enough to explain why the authors did not perform the analysis also on the initial wild material. The argument for the maternal effect is not valid in this case, in addition the effect of the host plants could be eliminated by using the same host in all three areas. By rearing the wild material in artificial conditions is an additional stress that can create bottleneck events. Was the number of insects per area the same to setup the F0 for rearing? If the number was too small or not equal among the three areas then this can have detrimental effects on gene expression and a bottleneck can be taken for granted. The artificial environment is itself a stress for the wild insects and the results shown can be the outcome of the artificial rearing and/or the temperature combined. In any case the authors need to explain this experimental setup and take into account in their analysis the effect of artificial rearing. This is a limitation of the study and should be extensively discussed.

Figure 1: it can go to the supplementary material

L108: do you mean 200 adults per area? Please specify

L140-145: please include the links to these databases

Figure 2: diagram is missing. Please include the correct one

L356-360: these are the instructions of the journal and should be deleted

L362-386: why is this part in bold?

As a general comment, the authors are not discussing the importance of identifying the genes involved in coping with temperature stress. How is climate change related to the invasive insects and their ability to cope with increased temperatures?

Round 2

Reviewer 4 Report

The authors should consult a native English speaker to proofread their manuscript. 

The A10 response is not satisfying. At the beginning of the discussion it is mentioned "this work can help to understand the response mechanisms to the environmental temperature of invasive species when they occupy different habitats, which is important in the context of the current climate change" but in their last response they claim that they are not interested in climate change. This is a controversy, mainly because invasive species are reinforced by climate change. I suggest a small modification in which the authors will stress out the connection between climate change and invasive species.